# Do Patients with Prostate Cancer Benefit from Exercise Interventions? A Systematic Review and Meta-Analysis

**DOI:** 10.3390/ijerph19020972

**Published:** 2022-01-15

**Authors:** Martin Færch Andersen, Julie Midtgaard, Eik Dybboe Bjerre

**Affiliations:** 1Department of Physiotherapy, University College of Northern Denmark, Selma Lagerløfs Vej 2, DK-9200 Aalborg, Denmark; 2Mental Health Centre Glostrup, University of Copenhagen, Nordstjernevej 41, DK-2600 Glostrup, Denmark; julie.midtgaard.klausen@regionh.dk; 3Department of Clinical Medicine, University of Copenhagen, Blegdamsvej 3B, DK-2200 Copenhagen, Denmark; 4The University Hospitals’ Centre for Health Research, Rigshospitalet, Ryesgade 27, DK-2200 Copenhagen, Denmark; Eik_bjerre@hotmail.com

**Keywords:** quality of life, metabolic health, prostate cancer, exercise, frailty, older adults, health

## Abstract

Men diagnosed and treated for prostate cancer experience severe adverse effects on quality of life (QoL) and metabolic health, some of which may be preventable or reversible with exercise, the benefits of which healthcare providers and patients increasingly acknowledge, though existing evidence on its effects varies in significance and magnitude. We aimed to review the effect of exercise on QoL and metabolic health in a broad prostate cancer population. A systematic search was conducted in nine databases and eligible trials were included in the meta-analytic procedure. All outcomes were stratified into aerobic exercise, resistance exercise, and a combination of both. The review identified 33 randomised controlled trials (2567 participants) eligible for inclusion. Exercise had a borderline small positive effect on cancer-specific QoL (standardised mean difference (SMD) = 0.10, 95% confidence interval (CI) −0.01–0.22), and a moderate to large effect on cardiovascular fitness (SMD = 0.46, 95% CI 0.34–0.59) with aerobic exercise being the superior modality (SMD = 0.60, 95% CI 0.29–0.90). A positive significant effect was seen in lower body strength, whole-body fat mass, general mental health, and blood pressure. No significant effect was seen in fatigue, lean body mass, and general physical health. We thereby conclude that exercise is effective in improving metabolic health in men diagnosed with prostate cancer, with aerobic exercise as the superior modality. The effect of exercise on QoL was small and not mediated by choice of exercise modality.

## 1. Introduction

According to the Global Cancer Observatory’s GLOBOCAN 2018 database [1], prostate cancer is the most frequent cancer type affecting men in Western nations, with an estimated 1.2 million incident cases annually worldwide. Progress in screening and treatment has resulted in increased survival rates, though not without severe adverse effects on patients’ physical and mental health, such as the increased risk of metabolic disease and reduced quality of life (QoL) [2,3]. Many of these adverse effects are treatment related. The disease stage and its risk of metastasis, combined with the patient’s overall condition, determine treatment options. When the disease is localised, patients are offered curative treatment, including surgery or radiotherapy. However, when the disease is metastatic at the time of diagnosis, patients are offered androgen-deprivation therapy (ADT), androgen receptor signalling inhibitors (ARSI) and chemotherapy, with palliative intent. It is estimated that almost 50% of patients will receive ADT or ARSI at one point during their treatment [4]. These men, especially those treated with ARSI, are more likely to develop metabolic conditions, such as type 2 diabetes, cardiovascular disease, abdominal obesity, or bone loss, potentially leading to osteopenia, osteoporosis, and bone fractures [5]. Furthermore, ADT is associated with a decline in QoL and sexual function, as well as the aggravation of fatigue [2]. Lastly, patients receiving ADT have a higher risk of other-cause mortality [6].

Some of these adverse effects may be preventable or reversible with exercise [7,8], which is increasingly recognised by healthcare providers and patients. Moreover, exercise has been found to improve patient-reported outcomes (PRO) such as QoL and fatigue [7,9,10] but also physiological outcomes such as cardiovascular fitness, muscle strength, and endurance [7,8,11,12] in men diagnosed with prostate cancer. Additionally, general physical activity has been associated with reduced overall and prostate cancer-specific mortality [13].

While various reviews investigating the effect of exercise on patients diagnosed with prostate cancer [7,8,9,10,11,12] have been published previously, these have focused exclusively on either PROs [10], cardiovascular health for patients on ADT [11], or the effect of resistance exercise specifically [12]. Only one comprehensive systematic review with a meta-analysis addressing the effect of exercise in men diagnosed with prostate cancer has been published previously. However, this was conducted in 2015 [7] and since then, the number of trials evaluating exercise interventions for prostate cancer patients has doubled. Therefore, we aimed to offer an updated systematic review examining the potential differential effect of specific exercise modalities in the broad population of prostate cancer patients, to support clinicians’ decision making when prescribing exercise.

The objective was to review and synthesise the effects of exercise interventions on QoL and metabolic health in men diagnosed with prostate cancer, and to examine possible differential effects across specific exercise modalities.

## 2. Materials and Methods

### 2.1. Protocol and Registration

This systematic review was conducted according to the Preferred Reporting Items for Systematic reviews and Meta-Analyses (PRISMA) (Appendix A). A protocol was registered on the International Prospective Register of Systematic Reviews, number CRD42020153851.

### 2.2. Eligibility Criteria

We included all randomised controlled trials (RCTs), allowing randomisation on both the individual and cluster level and feasibility trials. The following three criteria had to be met: (1) Participants with a history of clinically diagnosed prostate cancer regardless of tumour histology or tumour stage, and that had received cancer treatment. We did not include studies with a mixed cancer population (breast, lung etc); (2) Exercise, defined as “… a subset of physical activity that is planned, structured, and repetitive and has as a final or an intermediate objective the improvement or maintenance of physical fitness” [14], lasting at least eight weeks with components of aerobic and/or resistance exercise and reported details in relation to frequency, intensity, and duration. We defined aerobic exercise as any form of physical activity that produces an increased heart rate and respiratory volume to meet the oxygen requirements of the activated muscles [15]. Resistance exercise was defined as exercise that intends to improve or maintain the ability to create the most significant amount of power at a specific or prescheduled speed and type of muscle contraction [16]. Studies were included regardless of whether interventions were supervised or home-based and regardless of delivery method (individual, group, or mixed). Combinations with other types of interventions (dietary, psycho/social) were allowed; (3) Comparison groups receiving usual care or waiting-list protocols were included. We allowed comparisons with attention control groups, such as stretching protocols, but only objective outcomes were examined in these cases.

### 2.3. Search Strategy

A systematic search was conducted in June 2021. We identified literature from the following electronic databases: MEDLINE, EMBASE, Cochrane Central Register of Controlled Trials (CENTRAL), CINAHL, PsycINFO, SPORTDiscus, PEDroand Scopus^®^. Appendix A contains the complete MEDLINE search strategy. We also expanded the database search by scanning reference lists of included studies or relevant reviews identified through the systematic search. Possible relevant unpublished studies were searched for in the grey literature via the OpenGrey database. Lastly, we screened ClinicalTrials.gov (https://clinicaltrials.gov/, accessed on 11 August 2020) for ongoing trials.

### 2.4. Study Selection

Two authors (M.F.A. and E.D.B.) independently applied a three-step process to screen for eligibility: (1) screening of titles and abstracts to remove irrelevant studies; (2) full-text screening of potentially relevant studies to assess compliance with eligibility criteria; and (3) linking of multiple publications from the same trial by comparing author names, location, specific details, number of participants, date, and duration. A lack of consensus at all stages was resolved by discussion with a third author (J.M.).

### 2.5. Data Collection Process

Data were extracted on the following outcomes: cancer-specific QoL, cardiovascular fitness, cancer-related fatigue, general physical and mental health, lower body strength, lean body mass, whole-body fat mass, and blood pressure. M.F.A. extracted and entered data on an Excel (Microsoft Corporation, Redmond, WA, USA) spreadsheet that E.D.B. then independently checked for accuracy. If more than one publication was identified for each trial, we selected the first published publication as the primary reference.

### 2.6. Data Items

For all outcomes, if more than two time points (i.e., baseline and post-intervention) were reported, we selected what was closest to the end of the intervention for extraction.

### 2.7. Risk of Bias in Individual Studies

Risk of bias in individual studies was assessed using Cochrane’s risk of bias tool [17]. We excluded two domains, “blinding of the participants and healthcare providers” and “blinding of the outcome assessor of subjective outcomes” (self-reported), which were not relevant due to the nature of the intervention and because the participants were the assessors.

### 2.8. Summary of Measurements

Outcomes were presented as continuous values and effect estimates of individual trials calculated as either weighted mean difference (WMD) or standardised mean difference (SMD) with a 95% confidence interval (CI). WMD was used when an outcome was assessed with an identical instrument in all included trials, while SMD was used in cases with different assessment instruments measuring the same outcome between trials [18]. For trials presenting multiple intervention arms, all relevant interventions contributed to the meta-analysis. Each arm had a control group allocated by dividing the number of participants in the original control group into two equal groups, creating single pair-wise comparisons [18]. Post-intervention measurement and post-intervention standard deviation (SD) were used in the meta-analysis. If post-intervention SD was not reported, either baseline SD or a calculated SD from reported standard error (SE) was used.

### 2.9. Synthesis of Results

The meta-analytic procedure was conducted following a two-stage process. First, a summary statistic was calculated to describe the observed effect of the intervention in each trial. Second, a pooled effect estimate was calculated for each outcome using a random effect model which involves an assumption that the effect being estimated in different trials are not identical but follows a distribution.

Heterogeneity was tested using the chi-squared test, placing the significance level at 0.10 [18]. The quantity of heterogeneity was tested using the I-squared test [18]. Interpretation of the I-squared values was: 25% indicating low heterogeneity, 50% moderate heterogeneity, and 75% considerable heterogeneity [19].

### 2.10. Risk of Bias across Studies

To account for the possibility of publication bias a funnel plot was conducted for visual examination of asymmetry and statistically tested using Egger’s test [20], but these were only performed if a sufficient number of studies (>10) were available [21]. 

### 2.11. Additional Analyses

To explore potential heterogeneity, prespecified subgroup analyses were conducted, which involves stratifying the studies according to intervention modality, i.e., all outcomes were stratified into aerobic exercise, resistance exercise and a combination of the two. Furthermore, outcomes were stratified according to type of assessment instrument.

The subsequent sensitivity analyses were conducted to test the robustness of the results: change score values, random versus fixed effect, excluding cluster-randomised studies, excluding pilot/feasibility studies and excluding outliers. All analyses were performed using STATA version 15.1. (StataCorp LLC, College Station, TX, USA)

## 3. Results

### 3.1. Study Selection

A systematic literature search was conducted in August 2020. Of the 6693 identified records, 33 trials (2567 participants) were found to be eligible for inclusion in this systematic review [22,23,24,25,26,27,28,29,30,31,32,33,34,35,36,37,38,39,40,41,42,43,44,45,46,47,48,49,50,51,52,53,54]. We identified 24 linked publications with follow-up measurements or secondary analyses [55,56,57,58,59,60,61,62,63,64,65,66,67,68,69,70,71,72,73,74,75,76,77]. Figure 1 illustrates the study selection process.

### 3.2. Study Characteristics

Table 1 provides an alphabetical list of detailed within-study characteristics. While only RCTs were included in this systematic review, there was one cluster-randomised trial [39] and six randomised feasibility trials or pilot studies [24,25,27,29,30,35]. Patients included received different treatment modalities across and within the 33 included trials. Fifteen trials reported primarily patients on ADT treatment [20,22,23,24,25,27,30,38,40,42,44,45,46,48,49], six trials reported a combination of ADT and radiotherapy [28,29,32,33,39,43] and one trial reported radiotherapy alone [37]. Others reported previous prostatectomy [35,41], and previous surgery, radiotherapy or a combination of those [31]. Two trials reported patients undergoing active surveillance [26,34]. Three trials reported patients receiving various treatments such as radiotherapy, surgery, hormone therapy [51]; surgery, radiotherapy, ADT or a combination [36]; and watchful waiting, ADT or castration [19]. One trial did not report information on treatment [21].

Two trials compared two exercise intervention groups that met the eligibility criteria [31,47]. The patients with prostate cancer in the included trials varied both within and across trials regarding treatment modalities, tumour stage and aggressiveness. Likewise, the exercise interventions (e.g., frequency, duration, exercise modalities and combinations with other therapeutic components) varied across the trials: Bourke et al. [23] combined aerobic exercise and resistance exercise with dietary and behavioural therapy: Bourke et al. [24] combined aerobic exercise with behavioural therapy; Dieperink et al. [28] combined resistance exercise with pelvic floor exercises and nurse consulting; Eriksen et al. [29] and O’Neill et al. [43] combined aerobic exercise with dietary advice; Focht et al. [30] combined aerobic exercise and resistance exercise with dietary and cognitive therapy; and Hebert et al. [34] combined aerobic exercise with dietary and stress reduction therapy. One exercise arm in Taaffe et al. [47,48] and in Winters-Stone et al. [52] had a combination of resistance exercise and impact training, such as landing, jumping and skipping. Park et al. [44] combined resistance exercise with pelvic floor exercises. Mardani et al. [54] combined aerobic and resistance exercise with flexibility and pelvic floor exercise. Two studies, Uth et al. [49] and Bjerre et al. [22], investigated football interventions. Most trials used usual care or waiting-list comparator, while four trials used an attention control group. Assessment instruments for measuring outcome of interest varied across trials. For the outcome cancer-specific QoL, trials used either a prostate cancer-specific patient-reported outcome measure (PROM) (e.g., Functional Assessment of Cancer Therapy—Prostate) [22,23,27,38,40,41,43,45,46] or general cancer PROM (e.g., European Organization for Research and Treatment of Cancer Quality of Life Questionnaire-C30) [24,31,35,36,39,42,54]. For the outcome cardiovascular fitness, trials used either a maximum testing procedure (i.e., VO2peak/max) [23,24,29,37,38,40,41,46,49,51] or a submaximal walking test (i.e., 400 m walk test or 6 min walk test) [25,26,27,30,31,32,33,35,36,39,42,43,47,50]. 

### 3.3. Risk of Bias within Studies

Appendix A contains the risk of bias summary table, while the risk of bias with reason is available in Appendix A.

### 3.4. Synthesis of Results

Seventeen trials involving 1361 participants measured cancer-specific QoL [22,23,24,27,31,35,36,38,39,40,41,42,43,45,46,54]. Segal et al. [46] presented two intervention arms for comparison, providing a total of 16 comparisons eligible for meta-analysis. The meta-analysis showed a borderline significant pooled small effect estimate (SMD = 0.10, 95% CI −0.01–0.22) (Figure 2) in favour of exercise interventions and with no statistical heterogeneity (I-squared = 0.0%, *p* = 0.718). No significant difference was seen between exercise modalities, and no modality alone yielded significant effect estimates (Appendix A). Likewise, we found no difference between the assessment instruments (Appendix A). There was no indication of publication bias detected in the funnel plot or in the Egger’s test (Appendix A).

Twenty-four trials involving 1618 participants measured cardiovascular fitness [23,24,25,26,27,29,30,31,32,33,35,36,37,38,39,40,41,42,43,46,47,49,50,51]. Segal et al. [46] and Taaffe et al. [47] each presented two intervention arms for comparison. Meta-analysis was feasible for 22 trials, yielding 24 comparisons. Windsor et al. [50] only reported results on cardiovascular fitness after four weeks of intervention (not eight weeks, which was the end of the intervention). Focht et al. [30] reported only change scores, and post-intervention measurement could not be calculated as baseline measurements were not reported. Consequently, data from these studies were excluded from the meta-analysis.

Meta-analysis demonstrated an overall significant, moderate to large effect estimate in favour of exercise (SMD = 0.46, 95% CI 0.34–0.56) (Figure 3) and moderate heterogeneity (I-squared = 26,8%, *p* = 0.108). When comparing exercise modalities, all yielded a significant effect with aerobic exercise as the superior modality, corresponding to a large effect size (SMD = 0.60, 95% CI 0.29–0.90) (Appendix A). There was no difference between using direct measures of cardiovascular fitness (VO2 peak) or a submaximal measure (e.g., 400 m walk) (Appendix A). There was no indication for publication bias detected in the funnel plot or the Egger’s test (Appendix A).

Appendix A provides the forest plots for the remaining outcomes. Meta-analysis demonstrated a small significant overall effect of exercise on fatigue (SMD = 0.27 95% CI 0.13–0.40) (Appendix A). Two Hojan et al. [35,36] trials were excluded from the meta-analysis in relation to fatigue as they reported extreme effect estimates and the corresponding author did not respond when contacted about this. For the outcome lower body strength, a small to moderate overall effect was seen (SMD = 0.44, 95% CI 0.26–0.62) (Appendix A), while resistance exercise had a moderate to large effect (SMD = 0.66 95% CI 0.27–1.04) (Appendix A). Exercise showed no significant overall effect on lean body mass (WMD = 0.23 kg, 95% CI −0.51–0.97) (Appendix A); however, resistance exercise did yield a significant effect (WMD = 2.03 kg, 95% CI 0.30–3.75) (Appendix A). For the outcome whole-body fat mass, exercise demonstrated an overall significant reduction (WMD = −1.13 kg, 95% CI −1.92–−0.33) (Appendix A), with no difference between modalities (Appendix A). No effect was seen for exercise overall concerning general physical health (WMD = 1.40, 95% CI −0.58–3.38) (Appendix A); however, resistance exercise alone showed a significant effect (WMD = 2.61, 95% CI 0.44–4.79) (Appendix A). For general mental health, an overall effect was seen (WMD = 2.58, 95% CI 1.33–3.84) (Appendix A), with combined aerobic and resistance training as the superior modality (WMD = 3.64, 95% CI 1.33–5.94) (Appendix A). Lastly, a small reduction in both systolic and diastolic blood pressure was observed equal to −2.84 mmHg (95% CI: −6.34–0.67) (Appendix A) and −1.80 mmHg (95% CI −3.82–0.23) (Appendix A), respectively.

## 4. Discussion

### 4.1. Summary of Evidence

In this systematic review of RCTs reporting on the effect of exercise for men with prostate cancer, a borderline small positive effect was found on cancer-specific QoL, and a moderate to large positive effect was found on cardiovascular fitness. Furthermore, beneficial effects of exercise were found in the outcomes: lower body strength, whole-body fat mass, general mental health, systolic and diastolic blood pressure, and fatigue. However, no significant beneficial effects were identified on lean body mass and general physical health.

The beneficial effects of exercise on cancer-specific QoL are in line with a previous systematic review by Bourke et al. [7]; however, the magnitude of previous reported results was larger compared with this review. One possible explanation is that Bourke et al. based their conclusion on a sensitivity analysis that only included three studies assessed as high-quality studies. Our results do not provide evidence that one specific exercise modality is superior in improving cancer-specific QoL. This is supported by existing head-to-head evidence in two trials. Specifically, Santa Mina et al. [78], who compared aerobic exercise and resistance exercise, found no significant difference in health-related QoL in men with prostate cancer. Additionally, Segal et al. [46] found no significant effect of group assignment (aerobic exercise versus resistance exercise) on either prostate-specific QoL or cancer-specific QoL. Most of the trials in this systematic review reported no effects or small effects on cancer-specific QoL. Only one trial, by Bourke et al. [23], found significant and clinically relevant changes in favour of exercise when combining exercise with nutrition and behavioural support, suggesting that exercise alone may not be sufficient to improve cancer-specific QoL. Accordingly, researchers intending to improve QoL among men diagnosed with prostate cancer might investigate the potential mediating effect of combining exercise with other therapeutic interventions.

We found a small significant effect of exercise on fatigue, which is in line with findings from a recent review by Lopez et al. [10]. In contrast, a large systematic review with results that lacked statistical significance suggested that caution is warranted in terms of making any definitive, all-encompassing conclusions on the benefits of exercise on cancer-related fatigue [79]. Our results do not provide evidence that one specific exercise modality is superior in improving fatigue, which is supported by Taaffe et al. [47] who examined different exercise modalities on fatigue in men diagnosed with prostate cancer. Fatigue is believed to be the most frequent and influential debilitating symptom of prostate cancer and its treatment [80], and is a prevalent indicator of the QoL of cancer patients [81]. Thus, the lack of effect on fatigue may partially explain the modest effects documented in cancer-specific QoL in the current review. In addition, the modest changes in cancer-specific QoL and fatigue may also be due to floor and ceiling effects, as most studies did not aim to recruit participants with clinical levels of fatigue and/or suboptimal QoL. In this regard, a comprehensive individual patient data meta-analysis including 4519 cancer patients investigated the moderator effects of baseline values on various exercise outcomes [82] and found that the effect of exercise interventions post-cancer treatment on health-related QoL and fatigue appeared to be greater in patients with worse baseline values. Because no or marginal room for improvement exists in the case of high QoL and/or low fatigue at baseline, the beneficial effects of exercise may therefore be underestimated. Further research to investigate the effect of exercise in a target prostate cancer population reporting high fatigue and low QoL at baseline appears warranted.

The current meta-analysis found exercise to be useful in improving cardiovascular fitness and in lowering whole-body fat mass and blood pressure, which is in line with recent evidence on metabolic health in a prostate cancer population on ADT [11]. All exercise modalities were found to be effective in improving cardiovascular fitness with aerobic exercise as yielding large effect estimates. This is important in the light of evidence of a significantly increased risk of men with prostate cancer developing metabolic diseases, mostly due to ADT treatment [2]. In a recent study, cardiovascular fitness was inversely associated with the overall metabolic syndrome score [83]; hence, together with the improvements seen in the current study on isolated metabolic risk factors (i.e., blood pressure and body distribution), the benefits of exercise on cardiovascular fitness could be seen as highly clinically relevant for improving metabolic health in men with prostate cancer.

It is worth noting that the included studies assessed cardiovascular fitness by means of either a direct measure of maximum/peak oxygen uptake (most frequently the incremental cycling test while analysing gas exchange) or a sub-maximal performance test (most frequently the 400 m walk test or 6 min walk test). Most studies investigating aerobic exercise used a direct measure of maximum oxygen uptake [29,37,38,46,49], while studies investigating resistance exercise or a combination of resistance and aerobic exercise used a submaximal performance test. Such an increase in muscle strength in the lower body may have a positive effect on walking endurance without increasing oxygen uptake. However, submaximal walking tests (such as 6 min walk) have been found valid and reliable for evaluating cardiorespiratory fitness in patients with cancer [84]. Lastly, it must be acknowledged that improvements in metabolic health cannot be sustained without continuous adherence to exercise or physical activity, which is why it is imperative to conduct more studies examining and promoting the longer-term effects of exercise adoption in men diagnosed with prostate cancer.

The aim of the current systematic review was to examine possible differential effects across specific exercise modalities. We chose to stratify exercise interventions into aerobic exercise, resistance exercise, or a combination of the two. Our findings indicate that the choice of modality may be important when targeting physiological outcomes. Thus, not surprisingly, whereas aerobic exercise was superior in improving cardiovascular fitness (Appendix A), resistance exercise appeared superior in improving lower body strength (Appendix A) and lean body mass (Appendix A). However, exercise modalities did not show a mediating effect on PROMs such as cancer-specific QoL (Appendix A) and fatigue (Appendix A). This observation is supported by a position statement from the Exercise and Sports Science in Australia association, which concludes that the appropriate exercise prescription should be targeted and individualised according to patient- and cancer-specific considerations [85]. However, this is often a challenge in exercise RCTs because they follow a strict intervention protocol to ensure internal validity and to reduce the potential risk of bias. We encourage future research to allow a more flexible intervention protocol to investigate the effectiveness of individually tailored exercise according to physical needs and patient preferences.

### 4.2. Limitations

Some caution is warranted when interpreting the results of the current review. First, there was considerable within-study heterogeneity in the trial populations regarding cancer stage and treatment, which meant that conducting relevant subgroup analyses were not possible. Furthermore, the generalisability may be limited to patients treated with ADT, and likely those most motivated in terms of adherence to exercise, which may not be representative of the background population of men with prostate cancer. Caution is needed when transferring the results to men diagnosed with non-aggressive prostate cancer who are under active surveillance or with advanced prostate cancer with metastatic disease, as these groups are underrepresented in the current review. We encourage researchers to share raw data on patient characteristics (tumour histology, prior treatment, treatment response, and metastatic spread) to allow relevant subgroup analysis on individual patient’s data in future meta-analytic procedures. Second, even though exercise interventions were categorised into aerobic, resistance or combined exercise, the content varied across studies in relation to frequency, duration and intensity. We did not aim to investigate the moderating effects of these factors. However, most included trials used an exercise protocol lasting between 12 and 24 months with a frequency of two to three sessions per week [86]. This correlates with existing guidelines, which recommend a minimum of 12 weeks of combined aerobic and resistance exercise twice a week to meet the health challenges faced by prostate cancer treated with ADT. Future research may focus on the optimal frequency and intensity of exercise and how to best promote and support the long-term adoption of exercise behaviour in this population.

Furthermore, some studies combined exercise with other behavioural interventions such as dietary and cognitive behavioural therapy. The current review did not include trials investigating low-intensity exercise (e.g., qigong, tai chi [87] and yoga [88]), which may be a beneficial modality in improving QoL and fatigue. Furthermore, several trials reported no or insufficient information on adherence and fidelity, making it uncertain to what degree patients followed prescribed exercise. A recent systematic review stated that most studies investigating exercise interventions for prostate cancer survivors report adherence information on the frequency of exercise sessions completed but fail to report fidelity information on exercise intensity and duration [89].

The strengths of this systematic review and meta-analysis include a comprehensive search strategy, systematic risk of bias evaluation and a prespecified, systematic data synthesis.

## 5. Conclusions

In summary, the results of this comprehensive systematic review and meta-analysis suggest that men with prostate cancer can benefit from exercise. We found positive effects on metabolic health, specifically increased cardiovascular fitness, and reduced whole-body fat mass and blood pressure. Furthermore, exercise appeared to have a small positive effect on cancer-specific QoL and fatigue. These findings indicate that using exercise complementarily to prostate cancer treatment may mitigate the development of metabolic disease. Future research should focus on combining exercise with other behavioural interventions and/or should target patients with suboptimal baseline values who are deemed to be most likely to benefit from exercise. The choice of exercise modality may be important when targeting physiological outcomes; however, no modality appears to be superior when addressing PROMs.

## Figures and Tables

**Figure 1 ijerph-19-00972-f001:**
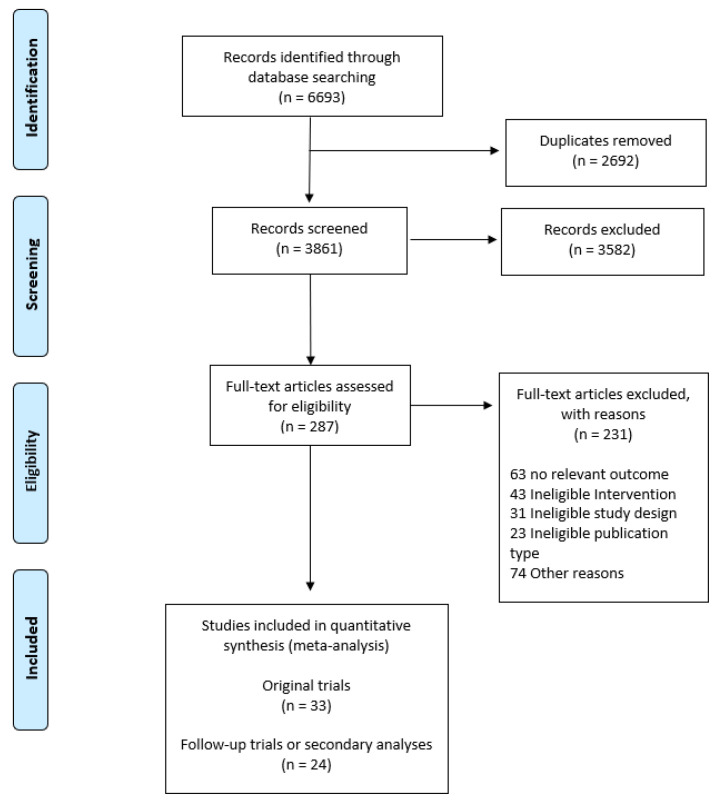
Flow diagram of literature and study selection process according to the 2009 Preferred Reporting Items of Systematic reviews and Meta-Analyses (PRISMA) with modification. Follow-up trials or secondary analyses represent reports involving the same study population for the same intervention trial.

**Figure 2 ijerph-19-00972-f002:**
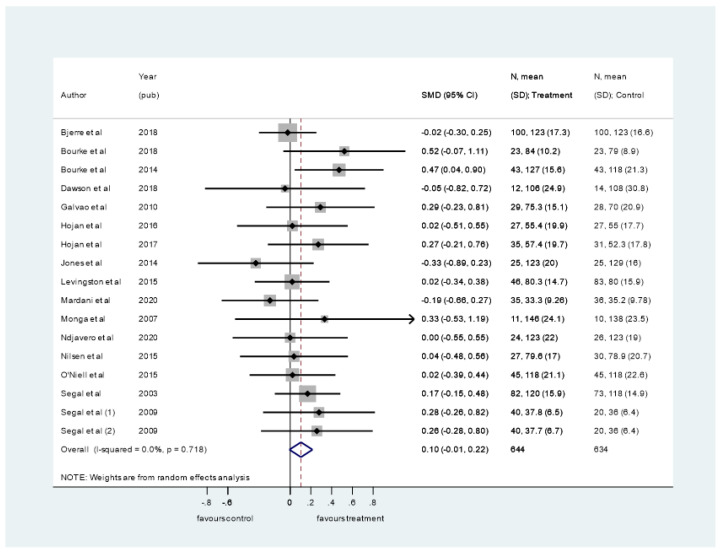
Pooled standard mean difference (SMD) on cancer-specific quality of life comparing exercise interventions with usual care or control in men with prostate cancer. A random effects model of DerSimonian and Laird, with estimate of heterogeneity from the Mantel–Haenszel model was used.

**Figure 3 ijerph-19-00972-f003:**
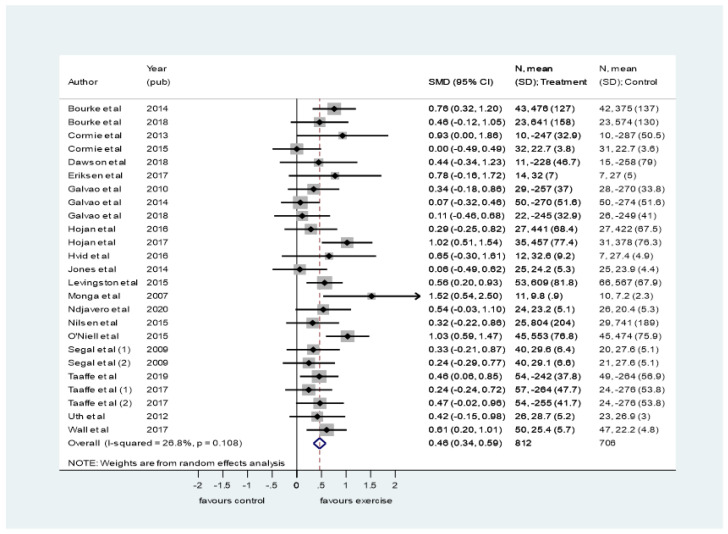
Pooled standard mean difference (SMD) on cardiovascular fitness comparing exercise interventions with usual care or control in men with prostate cancer. A random effects model of DerSimonian and Laird, with estimate of heterogeneity from the Mantel–Haenszel model, was used.

**Table 1 ijerph-19-00972-t001:** Study characteristics in alphabetic order.

Author (Year)	Sample (Group Randomised)	Population	Intervention Duration (Follow-Up)	Intervention Frequency	Intervention Description	Control Group	Outcomes of Interest
Bjerre et al., (2018) [22]	121 (109/105)	Tumour stage: Gleason mean 2.0–10.0 Metastatic disease: 19% PCa treatment: No treatment, ADT, castration	24 weeks	2 times per week	Supervised AT: 20 min of warm-up, 20 min of dribbling and shooting, 20 min of 5–7-a-side football	Usual care	QoL ^a^ Lean body mass(kg) ^b^ Body fat mass (kg) ^b^ General physical and mental health ^c^
Bourke et al., (2014) [23]	100 (50/50)	Tumour stage: T 3–4 Metastatic disease: 20% PCa treatment: ADT	12 weeks	First 6 weeks: 2 supervised, 1 home Last 6 weeks: 1 supervised, 2 home	Supervised AE: 30 min 55–75% max heartrate or Borg 11–13 RT: 2–4 sets 8–12 reps, 60% 1RM progressed through intervention Home AE + RT: 30 min of skills taught in supervised sessions	Usual care	QoL ^a^ Fatigue ^d^ Cardiovascular fitness ^e^ Systolic and diastolic BP
Bourke et al., (2018) [24]	50 (25/25)	Tumour stage: T 1–2 Metastatic disease: None PCa treatment: No information	12 weeks	2 times per week	Supervised AE: 20–30, 65–85% of age-specific heart rate max Home AE: 150 min moderate cardiovascular fitness per week	Usual care	QoL ^f^ Systolic and diastolic BP Cardiovascular fitness ^e^
Cormie et al., (2013) [25]	20 (10/10)	Tumour stage: Gleason mean 8.2 Metastatic disease: All (inclusion criteria) PCa treatment: Previous ADT	12 weeks	2 times per week	Supervised RT: 1–4 set (NI reps), 12–6 RM, 8 exercises Home AE: 150 min of moderate cardiovascular fitness per week	Waiting-list protocol	General physical and mental health ^g^ Fatigue ^h^ Cardiovascular fitness ^i^ Lower body strength ^j^ Lean body mass (kg) ^b^ Body fat mass (kg) ^b^
Cormie et al., (2015) [26]	63 (32/31)	Tumour stage: Gleason mean 7.5 Metastatic disease: None (excluded) PCa treatment: ADT	12 weeks	2 times per week	Supervised RT: 1–4 set (NI reps), 12–6 RM, 8 exercises AE: 20–30 min 70–85% of max heart rate Home AE: 150 min of moderate cardiovascular fitness per week	Waiting-list protocol	General physical and mental health ^g^ Fatigue ^k^ Cardiovascular fitness ^i^ Lower body strength ^k^ Lean body mass (kg) ^b^ Body fat mass (kg) ^b^ Systolic and diastolic BP
Dawson et al., (2018) [27]	37 (16/21)	Tumour stage: Gleason mean 7.5 Metastatic disease: 45% PCa treatment: ADT	12 weeks	3 times per week	Supervised RT: 3 sets 15–8 reps (decrease over time) 65–83% 1RM (increase over time) 10 exercises with alternatives	Stretching protocol 3 times a week	QoL ^a^ Fatigue ^m^ Cardiovascular fitness ^i^ Lower body strength ^l^ Lean body mass (kg) ^b^ Body fat mass (kg) ^b^ Systolic and diastolic BP
Dieperink et al., (2013) [28]	161 (79/82)	Tumour stage: T 1–3 Metastatic disease: No information PCa treatment: ADT	20 weeks	7 times per week	Home RT: Individualised programme, major muscle groups, 10–12 reps, 3 sets, moderate intensity 30 min per day	Usual care	General physical and mental health ^e^
Eriksen et al., (2017) [29]	26 (17/9)	Tumour stage: T 1–2 Metastatic disease: None PCa treatment: Active surveillance	24 weeks	3 times per week	Home AE: 45 min of 70% of max heart rate	Usual care	Cardiovascular fitness ^n^ Lean body mass (skinfold) Body fat mass (skinfold)
Focht et al., (2018) [30]	32 (16/16)	Tumour stage: Gleason mean 7.7 Metastatic disease: No information PCa treatment: ADT	12 weeks	2 times per week	Supervised RT: 3 sets, 8–12 reps (8–12 RM) 3–6 of perceived exertion (1–10), 9 exercises AE: 10–20 min 3–4 on perceived exertion PA guidelines (150 min per week)	Usual care	Cardiovascular fitness ^i^ Lower body strength ^j^ Lean body mass (kg) ^b^ Body fat mass (kg) ^b^
Galvao et al., (2010) [31]	57 (29/28)	Tumour stage: Gleason mean 7.2–7.5 Metastatic disease: 9% PCa treatment: ADT + RT	12 weeks	2 times per week	Supervised RT: 2–4 set (NI reps), 12–6 RM 6 exercises AE: 15–20 min 65–80% of max heart rate	Waiting-list protocol	QoL ^o^ General physical and mental health ^g^ Cardiovascular fitness ^i^ Lower body strength ^l^ Lean body mass (kg) ^b^ Body fat mass (kg) ^b^
Galvao et al., (2014) [32]	100 (50/50)	Tumour stage: T 2–4 Metastatic disease: None (excluded) PCa treatment: Previous ADT + RT	52 weeks	2 times per week	Supervised RT: 2–4 set (NI reps), 12–6 RM, 6 exercises AE: 15–20 min 65–80% of max heart rate Home AE + RT: After 6 months of replication of supervised programme	Information on physical activity with printed material	General physical and mental health ^g^ Cardiovascular fitness ^i^ Lower body strength ^j^ Lean body mass (kg) ^b^ Body fat mass (kg) ^b^ Systolic and diastolic BP
Galvao et al., (2018) [33]	57 (28/29)	Tumour stage: No information Metastatic disease: All (bone metastases, inclusion criteria) PCa treatment: ADT	12 weeks	3 times per week	Supervised RT: 3 sets, 10–12 RM major trunk, upper and lower body muscles, progressing 5–10% when exceeding RM values AE: 20–30 min at 60–85% of estimated heart rate max	Waiting-list protocol	General physical health ^c^ Cardiovascular fitness ^i^ Muscle strength ^j^ Lean body mass (kg) ^b^ Body fat mass (kg) ^b^
Hebert et al., (2012) [34]	54 (29/25)	Tumour stage: Unclear Metastatic disease: No information PCa treatment: Previous surgery, RT or both	24 weeks	5 times per week	Home AE: >30 min of moderate intensity exercise	Waiting-list protocol	Body fat mass (%) ^b^
Hojan et al., (2016) [35]	55 (27/28)	Tumour stage: Gleason mean 6.6 Metastatic disease: None (exclusion) PCa treatment: RT + ADT	8 weeks	5 times per week	Supervised RT: 2 sets, 8 reps, 5 exercises 70–75% of estimated 1RM AE: 30 min of moderate intensity, 65–70% of age-estimated heart rate max	Usual care	QoL ^o^ Fatigue ^d^ Cardiovascular fitness ^p^
Hojan et al., (2017) [36]	72 (36/36)	Tumour stage: Gleason mean 8.8 Metastatic disease: None (exclusion) PCa treatment: RT + ADT	52 weeks	5 times per week (weeks 1–10), 3 times per week (weeks 11–52)	Supervised RT: 2 sets, 8 reps, 5 exercises 70–75% of estimated 1RM AE: 30 min of moderate intensity, 65–70% of age-estimated heart rate max (weeks 11–52, 70–80% of estimated heart rate max)	Usual care	QoL ^o^ Fatigue ^d^ Cardiovascular fitness ^p^
Hvid et al., (2016) [37]	25 (12/7)	Tumour stage: T 1–2 Metastatic disease: None (exclusion) PCa treatment: Active surveillance	104 weeks	3 times per week	Home AE: 35 min of interval-based exercise varying between 50–100% of VO2max	Usual care	Cardiovascular fitness ^n^ Lean body mass (kg) ^b^ Body fat mass (kg) ^b^
Jones et al., (2014) [38]	50 (25/25)	Tumour stage: Gleason 62% >7.0 Metastatic disease: No information PCa treatment: Previous prostatectomy	24 weeks	5 times per week (supervised at least 3 times)	Supervised/Home AE: 30–45 min of walking with 55–100% of VO2peak	Usual care	QoL ^a^ Cardiovascular fitness ^n^ Lean body mass (%) ^b^ Body fat mass (%) ^b^ Systolic and diastolic BP
Livingston et al., (2015) [39]	147 (54/92)	Tumour stage: T 1–3 Metastatic disease: No information PCa treatment: Surgery, RT, ADT or combined	12 weeks	3 times per week (2 supervised, 1 home)	Supervised RT: 4–6 exercises, 2 sets, 8–12 reps AE: 20–30 min 40–70% of max heart rate Home RT + AE With bodyweight and resistance band	Usual care	QoL ^o^ Cardiovascular fitness ^p^ Lower body Strength ^l^ Systolic and diastolic BP
Mardani et al., (2020) [54]	80 (40/40)	Tumour stage: Unclear Metastatic disease: Unclear PCa treatment: Radiotherapy, surgery, hormone therapy	12 weeks	2 times per week	Supervised RT: 11 exercises for lang muscle groups, low to moderate load (11–13 on BORG), 8–12 reps, 2 sets AT: walk, light to moderate (11–13 on BORG), 60–150 min per week,	Usual Care	QoL ^o^ Fatigue t
Monga et al., (2007) [40]	(30) 21 (11/10)	Tumour stage: Gleason mean 5.3 Metastatic disease: Unclear PCa treatment: RT	8 weeks	3 times per week	Supervised AE: 30 min walking on treadmill, 65% of heart rate reserve	Usual care	QoL ^a^ Fatigue ^r^ Lower body strength ^s^ Cardiovascular fitness ^e^
Ndjavero et al., (2020) [41]	50 (24/26)	Tumour stage: Gleason 96% ≥7.0 Metastatic disease: 45% PCa treatment: ADT	12 weeks	2 times per week supervised, 3 times per week home	Supervised AE: 5 × 5 min on 11–15 on 55–85% of heart rate max on ergometer bike RT: Six exercises targeting major muscle groups, 2–4 sets, 10 reps at 11–15 RPE Home AE: 30 min self-directed structured physical activity	Waiting-list protocol	QoL ^a^ Fatigue ^k^ Cardiovascular fitness ^n^ Lean body mass (kg) ^y^ Body fat mass (kg) ^y^
Nilsen et al., (2015) [42]	58 (28/30)	Tumour stage: Intermediate high-risk profile Metastatic disease: Unclear PCa treatment: RT + ADT	18 weeks	3 times per week	Supervised RT: 9 exercises, Monday: 1–3 sets, 10 RM, Fridays: 2–3 sets, 6RM, Wednesdays: submaximal session, 10 reps, 80–90% of 10 RM, 2–3 sets	Usual care	QoL ^o^ Fatigue ^t^ Cardiovascular fitness ^u^ Lean body mass (kg) ^b^ Body fat mass (kg) ^b^ Lower body strength ^l^
O’Neill et al., (2015) [43]	94 (47/47)	Tumour stage: Gleason 90% ≥7.0 Metastatic disease: Unclear PCa treatment: ADT	24 weeks	5 times per week	Home AE: 30 min of brisk walking (moderate intensity)	Waiting-list protocol	QoL ^a^ Fatigue ^v^ Lean body mass (kg) ^b^ Body fat mass (kg) ^b^ Cardiovascular fitness ^p^
Park et al., (2012) [44]	66 (33/33)	Tumour stage: T 2–3 Metastatic disease: Unclear PCa treatment: Radical prostatectomy	12 weeks	2 times per week	Supervised RT: Elastic band, 50–70% of 1RM, upper extremities (lateral, anterior, posterior) lower extremities (lift and spread), weeks 9–12	Pelvic floor exercises	General physical and mental health ^g^
Segal et al., (2003) [45]	155 (82/73)	Tumour stage: T 2–4 Metastatic disease: No information PCa treatment: ADT	12 weeks	3 times per week	Supervised RT: 2 sets, 8–12 reps, 9 exercises, 60–70% of 1RM, 5 lbs increase when able to complete >12 reps	Waiting-list protocol	QoL ^a^ Fatigue ^d^
Segal et al., (2009) [46]	121 (40/40/41)	Tumour stage: T 1–4 Metastatic disease: None (excluded) PCa treatment: RT and ADT	24 weeks	3 times per week	Supervised AE: Weeks 1–4: 50–60% of VO2peak, weeks 5–24: 70–75%, duration 15 min progression 5 min every 3 weeks until 45 min RT: 2 sets, 8–12 reps of 10 exercises at 60–70% of 1RM	Waiting-list protocol	QoL ^a^ Fatigue ^d^ Cardiovascular fitness ^n^ Lower body strength ^w^ Body fat mass (%) ^b^
Taaffe et al., (2017) [47]	163 (58/54/51)	Tumour stage: Gleason mean 7.8 Metastatic disease: 7% of sample PCa treatment: ADT	24 weeks	RT + IMP: 2 times per week supervised, 2 times per week home (just IMP) RT + AE: 2 times per week	RT + IMP supervised RT: 2–4 sets (NI on reps), 6–12 RM, 6 exercises IMP: Ground reaction force 3,4–5,2 times body weight Home IMP: As supervised RT + AE supervised RT: 2–4 sets (NI on reps), 6–12 RM, 6 exercises AE: 20–30 min, 60–70% of estimated heart rate max Home AE: 150 min of moderate cardiovascular fitness per week	Waiting-list protocol	Fatigue ^t^ Cardiovascular fitness ^i^ Lower body strength ^l^ Lean body mass (kg) ^b^ Body fat mass (kg) ^b^
Taaffe et al., (2019) [48]	104 (54/50)	Tumour stage: Gleason mean 7.6 Metastatic disease: None, exclusion criteria PCa treatment: ADT	24 weeks	3 times per week supervised, 2 times per week home	Supervised AE: Treadmill, rower or bike 60–80% heart rate max RT: Major muscle groups 2–4 sets, 6–12 reps IMP: Hopping, skipping, leaping and drop jump (GRF 3.4–5.2 times body weight) Home Walking and modified IMP	Waiting-list protocol	Lean body mass (kg) ^b^ Body fat mass (kg) ^b^
Uth et al. (2014) [49]	57 (29/28)	Tumour stage: Gleason mean 7.9 Metastatic disease: Bone metastases 19% of sample PCa treatment: ADT	12 weeks	2 times per week (weeks 1–8), 3 times per week (weeks 9–12)	Supervised AE: 2 × 15 min of 5–7 a-side football (weeks 1–4), 3 × 15 min of 5–7 a-side football (weeks 5–12)	Waiting-list protocol	Lean body mass (kg) ^b^ Body fat mass (kg) ^b^ Cardiovascular fitness ^n^ Lower body strength ^j^
Windsor et al., (2004) [50]	66 (33/33)	Tumour stage: T 1–2 (51 out of 66) Metastatic disease: Unclear PCa treatment: RT	4 (8) weeks	3 times per week	Home AE: 30 min of walking at moderate intensity (60–70% of heart rate max)	Usual care	Fatigue ^m^
Wall et al., (2017) [51]	97 (50/47)	Tumour stage: Gleason mean 8.0 Metastatic disease: None (bone metastases excluded) PCa treatment: ADT	24 weeks	2 times per week	Supervised RT: 6 exercises, weeks 1–4: 2 sets, 12 reps, weeks 5–8: 3 sets, 10 reps, weeks 9–12: 3 sets, 8 reps, weeks 13–16: ? AE: 20–30 min, 70–90% of VO2peak Home 150 min per week 70–90% of VO2peak	Usual care	Cardiovascular fitness ^n^ Lean body mass (kg) ^b^ Body fat mass (kg) ^b^ Systolic and diastolic BP
Winters-Stone et al., (2014) [52]	51 (29/22)	Tumour stage: Unclear Metastatic disease: 27.6% of intervention sample, 13.6% of control sample PCa treatment: ADT	52 weeks	2 times per week supervised + 1 time per week home	Supervised RT: Lower body: 2–3 sets progression from 12–14 RM, 12–14 reps to 6–8 RM, 6–8 reps Upper body: 2–3 sets, 8–12 reps, progressing from 0–2% to 9–10% of body weight IMP: Jumps: 10 reps, progressing from 3 to 10 sets Home As supervised	Stretching protocol	Lean body mass (kg) ^b^ Body fat mass (kg) ^b^ Fatigue ^x^
Winters-Stone et al., (2016) [53]	64 (32/32)	Tumour stage: Unclear Metastatic disease: 9% of sample PCa treatment: Currently on ADT: Intervention (22%), control (13%)	24 weeks	2 times per week	Supervised RT exercising with spouse: 8–10 exercises, 8–15 reps, progressing from 15 to 8 RM	Waiting-list protocol	Lean body mass (kg) ^b^ Body fat mass (kg) ^b^ General physical and mental health ^g^ Lower body strength ^l^

The column “Intervention description” contains both supervised and home-delivered exercise characteristics. PCa, prostate cancer, AE, aerobic exercise, RT, resistance training, BP, blood pressure, QoL, quality of life, ADT, androgen-deprivation therapy, BW, body weight, IMP, impact training, NI, no information, RM, repetition maximum, reps, repetition; ^a^ FACT-P, Functional Assessment of Cancer Therapy—Prostate, ^b^ DXA, dual-energy X-ray absorption, ^c^ SF-12, 12-Item Short Form Health Survey, ^d^ FACT-F, Functional Assessment of Cancer Therapy—Fatigue, ^e^ Bruce ramp protocol, ^f^ EQ5D questionnaire, ^g^ SF-36, 36-Item Short Form Health Survey, ^h^ MFSI-SF, Multidimensional Fatigue Symptom Inventory-Short Form, ^i^ 400 m walk test, ^j^ 1RM leg extension, ^k^ FACIT-F, Functional Assessment of Chronic Illness Therapy—Fatigue, ^l^ 1RM leg press, ^m^ BFI, Brief Fatigue Inventory, ^n^ VO2max/peak (incremental cycling test with pulmonary gas exchange measurement), ^o^ EORCT QLQ-C30 (global), European Organization for Research and Treatment of Cancer Quality of Life Questionnaire-C30, ^p^ 6 min walk-test, ^r^ Piper Fatigue Scale, ^s^ Timed 5 reps of chair sit-to-stand, ^t^ EORCT QLQ-C30 (Fatigue), European Organization for Research and Treatment of Cancer Quality of Life Questionnaire-C30, ^u^ shuttle walk test, ^v^ FSS, Fatigue Severity Scale, ^w^ 8RM leg press, ^x^ Schwartz Cancer Fatigue Scale, and BIA, bioelectrical impedance analysis.

## Data Availability

Please contact the corresponding author to request data.

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
