# Peer review of "Do Patients with Prostate Cancer Benefit from Exercise Interventions? A Systematic Review and Meta-Analysis"

_ijerph, 2022, doi:10.3390/ijerph19020972_

Round 1
Reviewer 1 Report
Dear authors,
This manuscript performs an analysis of patients diagnosed with prostate cancer. Prostate cancer affects the quality of life of men. It is important to see if physical activity improves this quality of life. Therefore, this manuscript is very pertinent.
Methodologically, the manuscript is very well presented. The analysis carried out can be reproduced in the different databases and the results are illustrative.
Your manuscript is interesting but I need you to answer some questions:
MATERIALS AND METHODS
Search Strategy:
WOS is not a database, but a meta-search engine.
Meta-analysis:
The authors should describe well how the meta-analysis is carried out.
RESULTS
You cannot use the references of the "results" in "discussion". This is wrong. Authors should look for alternative references to justify their "results".
REFERENCES
Some references are incomplete or have errors. References must be unified. For example, authors sometimes cite the DOI and sometimes they cite the web. The authors should review this section.
Author Response
Dear reviewer
Thank you for your comment, which we believe will improve our manuscript. Here is our point-to-point response to your comments.
Methods
We agree that Web of Science is a meta-search engine and have removed it from our manuscript.
We have now added an adequate description of our meta-analytical procedure
Results
We are not sure how to handle the comment from the reviewer regarding the use of the reference "results" in "discussion"? We wish to accommodate the reviewer's comment; however, is it possible to elaborate the exact correction suggested to be made?
Reference
All references have been reversed and aligned with DOI
Reviewer 2 Report
The manuscript is well-written and the meta-analysis is carried out rigoursly. I appreciate the effort made by the authors to conduct this study. However, some minors questions should be addressed by the authors before the manuscript can be considered for publication.
Introduction
- It is necessary to justify deeply the necessity of carried out a similar systematic review in the field.
Method
- The flowchart has been carried out the PRISMA guidelines. The authors should add an statement to specify that the systematic review and the meta-analysis was carried out following the PRISMA guidelines. Moreover, it is necessary to apply the PRISMA checklist and add this document as a supplementary file.
- As the authors state, the selection and codification of the studies have been carried out by two coders independently. It the authors have data, some inter-judge reliability coefficient should be computed.
Author Response
Dear reviewer
We thank you for your comments, which we believe will improve our manuscript. Please see the point-to-point response.
We have revised the section explaining the necessity of this broard systematic review examing different exercise modalities. We hope it will be satisfactory.
We have added the following sentence to the manuscript "This systematic review was conducted according to the Preferred Reporting Items for Systematic reviews and Meta-Analyses (PRISMA)."
A PRISMA checklist has been added to the supplementary file (ESM_6)
Two authors conducted the study selection process and risk of bias assessment independently. Unfortunately, we do not have sufficient data to run an inter-judge reliability coefficient.